# Right Knee—The Weakest Point of the Best Ultramarathon Runners of the World? A Case Study

**DOI:** 10.3390/ijerph17165955

**Published:** 2020-08-17

**Authors:** Robert Gajda, Paweł Walasek, Maciej Jarmuszewski

**Affiliations:** 1Center for Sports Cardiology, Gajda-Med Medical Center in Pułtusk, 06-100 Pułtusk, Poland; 2Traumatology and Orthopedic Department, Bielanski City Hospital, 01-809 Warszawa, Poland; dr.walasek@gmail.com; 3Luxmed Diagnostic, 01-809 Warszawa, Poland; maciej.jarmuszewski@luxmed-diagnostyka.pl

**Keywords:** clockwise run, counter-clockwise run, 24-h ultramarathon run, knee injury, endurance sports, 24-h UM

## Abstract

The impact of ultramarathons (UM) on the organs, especially in professional athletes, is poorly understood. We tested a 36-year-old UM male runner before and after winning a 24-h marathon. The primary goal of the study was cardiovascular assessment. The athlete experienced right knee pain for the first time after 12 h of running (approximately 130 km), which intensified, affecting his performance. The competitors ran on a 1984 m rectangle-loop (950 × 42 m) in an atypical clockwise fashion. The winner completed 516 rectangular corners. Right knee Magnetic Resonance Imaging (MRI) one day after the run showed general overload in addition to degenerative as well as specific features associated with “turning to the right”. Re-examination after three years revealed none of these findings. Different kinds of overloading of the right lower limb, including right knee pain, were indicated in 6 of 10 competitors from the top 20, including a woman who set the world record. The affected competitors suggested as cause for discomfort the shape of the loop and running direction. They believed that changing the direction of the run during the competition and an athletics stadium loop shape on a 2000–2500 m length is better for 24-h UM runners. In the absence of technical alternatives, the “necessary evil” is a counterclockwise run (also Association of Athletics Federations IAAF recommendation). Results suggest that a one-way, clockwise, 24-h UM run had an adverse effect on the athlete’s right knee, as a result of unsymmetrical load. Organizers of 24-h UM runs should consider the shape of the competition loop and apply the principle of uniform load on the musculoskeletal system (alternate directions run). In case of technical impossibility, it would be better to run counterclockwise, which is more common, preferred by runners, and recommended by the IAAF.

## 1. Introduction

Ultramarathon (UM) races (distance longer than 41,195 m) are a significant challenge for the competitors’ organism. Preparations for these races take years, and often crown a runner’s career. Overload injuries of the musculoskeletal system due to grueling training is highly probable, and often results in the inability to start the race for which the athlete has prepared for years. It is not uncommon for an overload injury to end a runner’s career. Competitors can train to minimize the probability of injury using various training stimuli, rehabilitation, or avoiding repetitive movements. This special attention to training may be insufficient for the safe completion of a 24-h (24H) race if it takes place, for example, on a rectangular loop and in one direction. This was the case during the 24H race in the Polish Championships. Six of the top 10 competitors reported right knee pain, probably due to taking several hundred rectangular turns to the right. Two of the world’s leading UM runners participated in the run, and both experienced similar pain. Fortunately, their overload injuries did not end their careers. However, such a situation requires an assessment of the impact of such organized races on the competitors’ health, and indicates the necessity and impact of this study.

Conducting sport competitions, especially long street runs, is a challenge for organizers, both because of the need to ensure the accuracy and indisputability of the results, and also because of competitors’ safety. In the case of 24-h runs, there is also a need for accommodation for teams taking care of competitors (coaches, masseurs, physiotherapists, chefs, and others) and the audience itself. The best competitors in the 24-h UM run distances of 250 to 300 km [1]. Since time but not distance is predetermined and it is impossible to predict where competitors will finish the race, usually such competitions take place on loops of different lengths and shapes. These loops have a relatively small range of lengths: from a typical athletics track of 400 m to 1–2 miles [1]. It depends not only on organizational capabilities but also on athletes’ preferences and the need for frequent technical support (preferably at least every 2.5 km). The mental sense of security and the ease of controlling the competition course on a shorter loop are also relevant. Different sports competitions have competitors running in different directions. Often, the organizers change the running direction, for example in the Taipei 24-h UM run (treadmill 400 m, changing the running direction every 4 h) [2]. However, loop shapes are mainly dependent on technical capabilities, from extremely uncomfortable with turns at a 180-degree angle, to routes running on different surfaces (dirt ground) with slight slopes [3]. Organizers often forget that loop shape not only affects the final score but also the athlete’s health during the run and regeneration afterwards. To date, there are not many reports on the impact of UM on athletes’ knees, and none describing the impact of running direction and loop shape. Our case study is an example of a leading UM world runner, who during the competition struggled not only with rivals but also with progressive pain in the right knee. Because the athlete did not use heart rate monitors during the 24-h UM, we could not judge how the growing pain affected his speed and or his intensity of effort at each stage of the competition [4,5]. Similar discomfort has been reported by a number of leading athletes, which inspired us to conduct an interview regarding not only pain after the run but also technical preferences (loop length and running direction) for 24-h UM runs.

## 2. Materials and Methods

### 2.1. Sports Biography and Main Achievements

The UM runner was 36 years old on the day of the competition (height: 1.73 m, weight: 63 kg, body mass index: 21.05 kg/m^2^). Despite a sedentary job working Monday to Friday from 8:00 h to 16:00 h, he had been running regularly for 20 years and had run approximately 100,000 km. For the last year, his daily training consisted of an average of 22 km/day from Monday to Friday and about 37 km on weekend days. He also swims and goes to the gym for cross-training. This athlete has participated in approximately 50 UMs, the longest of which was a 48-h run. So far, he has never been injured or seriously ill. A two-time Polish Champion in the 24-h UM and winner of the Spartathlon in Greece (2016, 246 km), he also holds numerous other honors. In 2014, he set the Polish record for 12-h races (145.572 km) and has participated four times in the 24-h UM World Championships, winning a medal each time. In 2019, he won the 48-h race in Athens (362 km).

All his achievements are on the official website of Ultra Marathon Statistics [6].

### 2.2. Anamnesis and Information from Other Competitors

We collated information on post-race pain and preferred lengths, loop shapes, and running direction from 10 athletes in the top twenty of the 24-h UM run who freely agreed on providing such information either directly after the race, or via subsequent e-mail correspondence and telephone conversations.

### 2.3. Methods

The athlete tapered his training (i.e., gradually reduced exercise over a short period of time and then stopped completely before the competition) two weeks before start of the 24-h UM. The run started at 12:00 AM on 8 April 2017. The competitors ran on a 1984 m asphalt loop in a clockwise fashion. The loop was a 950 m by 42 m rectangle (Figure 1 and Figure 2). Our study subject ran 129 loops and performed 516 rectangular turns. On one of the straight sections, there was a tent in which the athlete could stop for a meal, a short rest, and basic hygiene. After 12 h, he stopped for 12–15 min for a warm meal, a change of shoes, and other hygienic activities. In addition, he used the toilet three times (2–3 min each). He ate and ingested liquids along the run. The weather conditions are presented in Table 1.

#### 2.3.1. Study Protocol—MRI

MRI was performed twice: one day after the 24-h UM run and exactly 3 years later, using a Philips Achieva 3T TX clinical scanner (Philips Medical Systems, Aurora, OH, USA) with a knee extremity coil. The patient underwent imaging according to the standard knee protocol of two-dimensional (2D) thin slices in sagittal (PDW/PDW SPAIR), coronal (PDW SPAIR/PDW TSE), paracoronal ([ACL]: T2W TSE) and axial (T1/T2FS/PDW SPAIR) planes. MRI acquisition of isotropic three-dimensional images (3D) fast spin-echo sequences (sagittal: PD SPAIR HR/HI-3D) was performed as well. The menisci, ligaments, and cartilage were assessed using the standard sequence.

#### 2.3.2. Ethical Approval

This case report was approved by the ethical review board of the Bioethics Committee of the Healthy Life Style Foundation in Pułtusk (EC 3/2017/medicine/sports, approval date: 30 March 2017). All runners provided their written informed consent to participate in the analysis and for their data to be published.

## 3. Results

### 3.1. Data Related to 24-h UM Run

During the 24-h UM, the athlete ran 258.228 km. He began the UM without any symptoms (no pain). The athlete started to experience right knee pain after approximately 12 h of running (approximately 130 km), which intensified until the end of the race. He ran on a 1984 m rectangular loop (950 × 42 m). The winner completed 516 rectangular corners, running in an atypical clockwise fashion. The women’s world record in the 24H UM was set in the same race with a result of 256.245 km.

### 3.2. MRI Was Performed Twice: 1 Day after the 24-h UM Run and Exactly 3 Years Later

(a)Right knee MRI: One day after 24-h UM run

MRI examination revealed considerable joint effusion with synovial edema. Linear tear of the anterior root of lateral meniscus was diagnosed, whereas the medial meniscus was intact. Both crucial ligaments were intact. There were overload and post-traumatic lesions tangential to the distal biceps femoris short head and around the plate of the myotendinous junction. We also observed an edema of distal muscle fibers of biceps femoris and a small hemolyzed hematoma of this area penetrating tangentially to the posterior medial edge of the fibula head. There were features of lateral collateral ligament (LCL) overload changes expressed by distal ligament edema and periarticular soft-tissue edema.

On the other hand, the medial collateral ligament (MCL) was intact. The quadriceps and patellar tendons were normal. The width of the articular cartilage of the tibiofemoral and patellofemoral joints were normal with a homogenous signal, no signs of chondromalacia, and usual subchondral bone marrow signal. The joint space was identified as normal. There was a trace of fluid in the semimembranosus-gastrocnemius bursa and an edema in the Hoffa’s fat pad.

(b)Right knee MRI: three years later

There was a large effusion with synovial edema in the joint cavity. We observed a progression of lateral meniscal tear. We detected a horizontal, stable tear of the lateral meniscal body, without dislocation. The medial meniscus was intact. We observed no changes in either crucial or collateral ligaments. The quadriceps and patellar tendons were normal. The appearance of articular cartilage of tibiofemoral and patellofemoral joints showed no abnormalities. We observed a normal subchondral bone marrow signal. There was no overload periarticular edema of soft tissue. We detected traces of fluid in the semimembranosus-gastrocnemius bursa and the edema Hoffa’s fat pad.

### 3.3. Interview Regarding Pain, Preferred Running Direction, Shape, and Loop Length for 24-h UM Run

We gathered information on post-race pain, preferred loop shapes, and preferred running direction from 10 UM runners from the top 20 of the 24-h UM run. The predominant ailment was right knee pain (50%), followed by right hip pain (20%), and left calf pain (20%). Their preferences were treadmill shape: athletic stadium-shape (70%), loop length: 2000 or 2500 m (90%), and running direction: counterclockwise (100%). Detailed results are presented in Table 2.

## 4. Discussion

### 4.1. Discussion of the Results

The right knee pain reported by the athlete after the run was cause for MRI examination. MRI results, in addition to degenerative changes, showed asymmetrical changes associated with overload of the lateral compartment of the right knee joint. The runner completed 516 rectangular corners. During the run, his right knee was exposed to centrifugal force. This force affected the lateral knee compartment, leading to pain after the run. Posterolateral knee stabilization is provided by the arcuate ligament complex, comprising of the lateral collateral ligament, the biceps femoris tendon, the popliteus muscle and tendon, the popliteal meniscal and popliteal fibular ligaments, the oblique popliteal, the arcuate and fabellofibular ligaments, and the lateral gastrocnemius muscle [6]. The most important active stabilizer of the lateral side of this joint is the biceps femoris, and its distal attachment to the fibula head. Passive stabilization is provided by the lateral collateral ligament [7]. Both structures appeared overloaded in MRI images. MRI imaging revealed edema and overload changes in the attachment distal biceps femoris tendon to the fibula head, a small hemolyzed hematoma penetrating tangentially to the posterior medial edge of the fibula head, features of LCL overload, distal ligament edema, linear tear of the anterior root of the lateral meniscus, and periarticular soft-tissue edema (Figure 3A,B). It is highly likely that the lateral knee compartment overload caused by completion of over five hundred rectangular corners could be the reason for the described MRI observations. Most changes (except for the lateral meniscus tear, which cannot heal) were not observed after three years.

The interview collected data from 10 out of 20 leading competitors to determine pain classes after the run and their preferences for this type of competition. Half of them, including two world-class runners, reported pain in their right knee. According to UM runners, the ideal loop shape for a 24-h UM run should resemble a typical 400 m stadium track and have a length of 2000 or 2500 m. The race should take place in alternating directions. In situations of technical impossibility, the widely preferred direction is counterclockwise. The athlete’s knee pain can be explained by the overload associated with the unfortunate rectangular shape of the loop (four right angles on each loop) and in addition, the opposite direction of the run was a problem for the athlete, who had adapted to “left turns” after years of training. The reasons for a preferred “counterclockwise” direction, otherwise generally recognized as “natural”, are a matter of speculation (“heart on the left, right leg stronger and longer, etc.”) rather than scientific evidence (see discussion below).

### 4.2. Knee Overload Injuries in UM Runners

In longer ultramarathons, approximately 50–60% of participants experience musculoskeletal problems. Most common injuries in ultramarathoners involve the lower limb, e.g., the ankle or the knee [8,9,10]. The knee is the most common region of exercise-related injury [11], accounting for 20% of all injuries among 161 km ultramarathon runners [12].

Some MRI studies show that even running 4487 km in a multistage UM does not cause permanent negative changes in the knee joint [13,14]. Potential overload injuries during long runs are influenced by a number of individual factors: biochemical, clinical, and anthropometric [15]. There are a few well-documented studies that testify knee joint adaptability, even on extreme mountain UMs [16]. A prospective MRI study of one knee from ten randomly selected participants who completed the Comrades Marathon between 1997 and 2002 was performed by Hagemann et al. UM runners’ knees were scanned 48 h before the race, and 48 h and one month after the 89 km oldest UM race. The race appears to have had a detrimental effect on runners who started the ultramarathon with tendinopathy, which worsened post-race by MRI criteria. One month after the race, the scan appearance of the injury had either improved or resolved completely [17]. None of these studies, however, looked at the tendency of just one knee (e.g., right) to undergo more frequent stress injuries. To our knowledge, there are no previous studies on 24-h runs in a single direction and related asymmetrical knee overload.

### 4.3. Why Do Athletes Run Around the Track Counterclockwise?—A Lot of Gossip and Poor Scientific Evidence

Although obvious for many, it is not scientifically proven. There are many popular scientific theories on this subject. According to Tavakkoli and Jose:
“In 1896, the First Modern Olympic Games was held in Athens, Greece. During this event the athletes were required to run clockwise during the track events. This was met with much complaint from the athletes. It was because of these complaints that the IOC then gathered in 1913 and set the current anticlockwise rule. We run counterclockwise because everything in nature tends towards counterclockwise motion. An spectator will perceive the runners as moving left to right, the same direction our eyes move when we read. The human body is slightly heavier on the left than the right because of the heart and when running anticlockwise, the body would tend to very slightly incline towards the left, which could be an advantage while running anticlockwise. Because most people are right hand/leg dominant, moving counterclockwise we have a better control and move faster.”[18]

Ultimately, studies by Bestaven and others could not confirm any of these theories. He studied many aspects responsible for the tendency to turn. For instance, with closed eyes people turned left (50% of respondents), right (39%), or kept on moving straight (11%). These studies confirmed that veering is not due to mechanical asymmetries but could be related to an asymmetry in sensorial inputs [19]. Regardless, just because 50% of people with eyes closed turn left does not mean running left is healthier than running right. Most sports, and in fact most other things involving some sort of circular movement, do go counterclockwise. Think about athletics track races, track bicycle races, speed skating, roller derby, the customary flow of people around an ice-skating rink, the direction of dancers moving around a dance floor, motorcycle speedway, NASCAR racing, horse racing, greyhound racing [20]. Other reasons (“better to the left direction”) are more suggestions, and are not necessarily scientific or proven facts [20]. The guidelines from the International Association of Athletics Federations (IAAF) on this topic agree with the counterclockwise preference of the athletes in our study and state: “The direction of running shall be left-hand inside” (IAAF Rule 163.1) [21].

### 4.4. Limitation

An important limitation is the fact that MRI examination of one knee was performed right after the run. We do not have a pre-run test or second knee for comparison; neither do we have other athletes in the sample. Unfortunately, the individual biomechanical conditions of the runner were not taken into account in the assessment of knee loads. Follow-up examination after three years is a distant examination. An explanation is the fact that the planned extensive initial and control tests of the athlete had a cardiological profile [22]. Orthopedic complaints reported at the finish line by the examined athlete and other participants became the reason for performing the knee examination and subsequent interviews. Right knee MRI follow-up was accompanied by extensive cardiological re-diagnosis and performed as a preliminary pre-competition test (which did not take place due to the COVID-19 pandemic).

### 4.5. Perspectives

To properly document the harmful effects of running in one direction in a 24-h UM run, MRI tests of both knee joints on many competitors should be performed both before and after the run. Ideally, the study should be conducted in three different 24-h UMs conducted in clockwise, counterclockwise, and alternating directions. Such results could then provide a strong argument for the competition organizers of 24-h UM to consider loop characteristics and competition methodology.

## 5. Conclusions

Our results suggest that a one-way, clockwise, 24-h UM run had an adverse effect on the athlete’s right knee as a result of unsymmetrical load which distorted the final result. Organizers of 24-h UMs, responsible for the competitors’ health, objectivity, and results quality, should also take into account the shape of the loop and apply the principle of uniform load on the musculoskeletal system performing 24-h UM in alternate directions. In a situation of technical impossibility, it would be better to run in a counterclockwise direction, which is more common, preferred by runners, and recommended by the IAAF.

## Figures and Tables

**Figure 1 ijerph-17-05955-f001:**
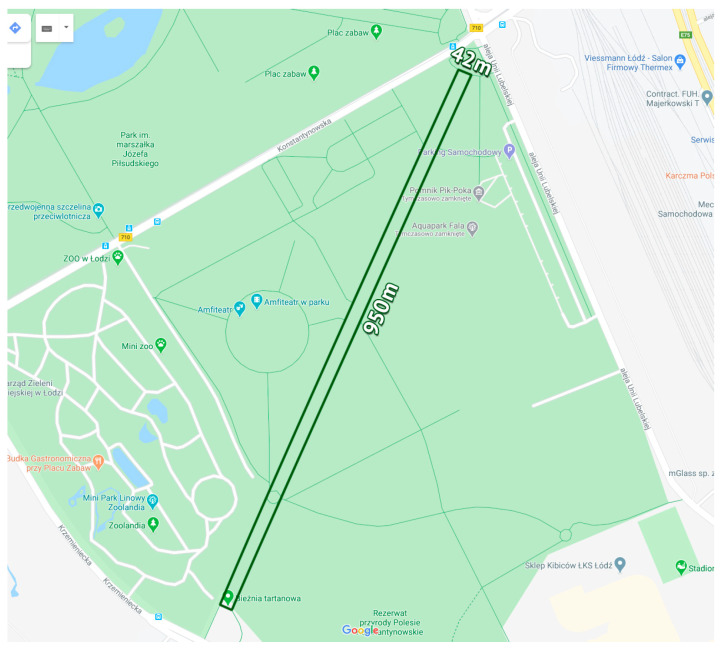
The loop on which 24-h ultramarathon run took place (Google Maps).

**Figure 2 ijerph-17-05955-f002:**
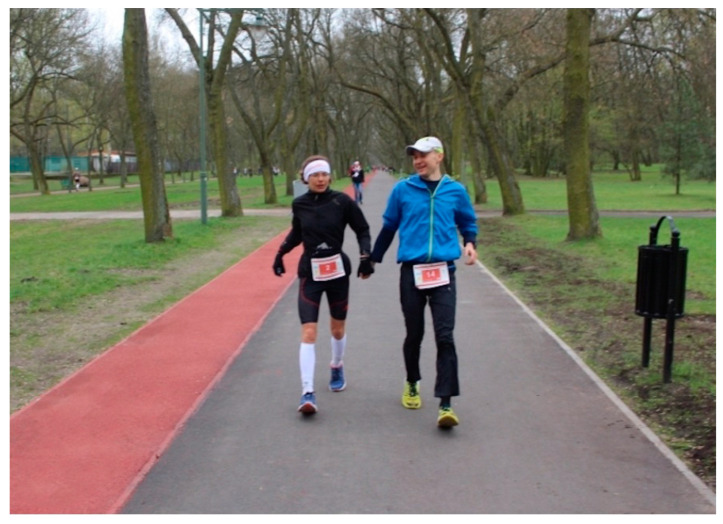
Asphalt surface on which the race took place. In the photo: winners of the men’s and women’s categories just before the final whistle.

**Figure 3 ijerph-17-05955-f003:**
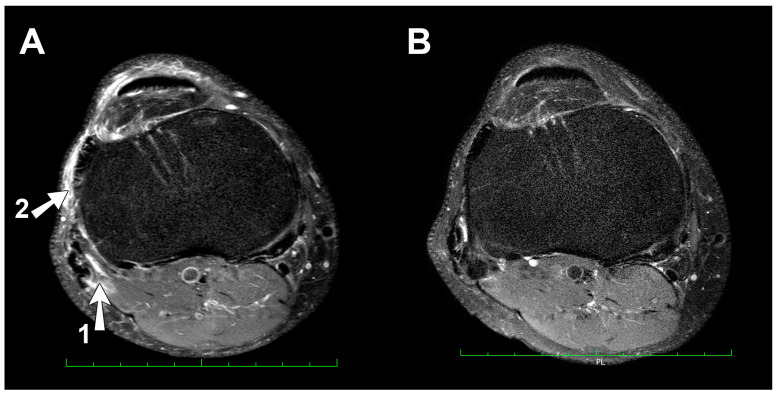
(**A**) Magnetic Resonance Imaging 2017 (axial scan), overload and post-traumatic lesions tangential to the distal biceps femoris short head and around the plate of the myotendinous junction (arrow 1), an edema, a small hemolyzed hematoma penetrating tangentially to the posterior medial edge of the fibula head and periarticular soft-tissue edema (arrow 2). (**B**) Re-examination in 2020. Overload effects from post-run examination are absent.

**Table 1 ijerph-17-05955-t001:** Detailed weather conditions in Łódź, Poland, during a 24-h UM competition according to Weather by CustomWeather, © 2020 8–9 April 2017 [7].

Time	Conditions	Comfort	Barometer	Visibility
	Temp	Weather	Wind	Humidity
12:00	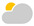	10 °C	Partly sunny	7 km/h	71%	1021 mbar	20 km
13:00	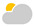	12 °C	Partly sunny	9 km/h	64%	1021 mbar	20 km
14:00	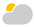	13 °C	Partly sunny	9 km/h	58%	1021 mbar	20 km
15:00	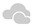	13 °C	Overcast	9 km/h	59%	1021 mbar	20 km
16:00	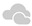	13 °C	Overcast	7 km/h	59%	1021 mbar	20 km
17:00	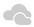	12 °C	Overcast	7 km/h	67%	1022 mbar	20 km
18:00	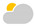	11 °C	Partly sunny	11 km/h	74%	1022 mbar	16 km
19:00	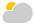	10 °C	Partly sunny	9 km/h	80%	1023 mbar	11 km
20:00	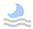	9 °C	Fog	7 km/h	85%	1024 mbar	8 km
21:00	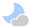	8 °C	Overcast	6 km/h	86%	1025 mbar	10 km
22:00	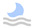	7 °C	Fog	2 km/h	90%	1025 mbar	7 km
23:00	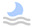	6 °C	Fog	4 km/h	93%	1025 mbar	4 km
00:00	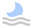	4 °C	Fog	2 km/h	93%	1026 mbar	2 km
01:00	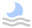	2 °C	Fog	2 km/h	96%	1026 mbar	1 km
02:00	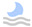	2 °C	Fog	No wind	97%	1026 mbar	0 km
03:00	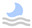	0 °C	Fog	No wind	98%	1026 mbar	0 km
04:00	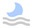	2 °C	Fog	2 km/h	99%	1026 mbar	0 km
05:00	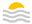	3 °C	Fog	2 km/h	99%	1026 mbar	0 km
06:00	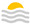	3 °C	Fog	No wind	99%	1027 mbar	1 km
07:00	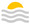	4 °C	Fog	N/A	100%	1027 mbar	1 km
08:00	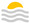	5 °C	Fog	N/A	100%	1027 mbar	2 km
09:00	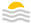	6 °C	Fog	No wind	100%	1027 mbar	3 km
10:00	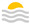	8 °C	Fog	2 km/h	96%	1028 mbar	8 km
11:00	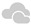	8 °C	Overcast	4 km/h	87%	1028 mbar	11 km

**Table 2 ijerph-17-05955-t002:** Interview data from 10 competitors from the top 20 24-h UM runners regarding pain, preferred loop shapes, and running direction.

Nr.	Distance Run during 24-h UM > 200 km	Sex	Right Leg Pain	Left Leg Pain	Preferred Loop Shape	Preferred Loop Length	Preferred Direction of Running—If Possible to Choose	Preferred Direction of Running—If Only One Way
1	yes	m	Knee	no	athletic stadium-shape	2000 m	Alternately	Counterclockwise
2	yes	f	Knee	no	athletic stadium-shape	2000 m	Alternately	Counterclockwise
3	yes	m	no	no	athletic stadium-shape	2500 m	Alternately	Counterclockwise
4	yes	f	Knee and hip	Achilles pain	athletic stadium-shape	2500 m	Alternately	Counterclockwise
5	yes	m	no	no	∞-shape	2000 m	Alternately	
6	no	m	no	ankle joint, calf strain	athletic stadium-shape	2500 m	Alternately	Counterclockwise
7	no	f	Iliotibial band issue	no	athletic stadium-shape	2000 m	Alternately	Counterclockwise
8	no	m	Knee and Calf strain	no	∞-shape	2000 m	Alternately	Counterclockwise
9	no	f	no	no	∞-shape	2500 m	Alternately	Counterclockwise
10	no	m	Knee and hip	Calf strain	athletic stadium-shape	1000 m	Alternately	Counterclockwise

f—female; m—male. ∞-shape—Competitors run around a loop shaped like ∞ (infinity sign). Continuing to run in one direction, they make one turn clockwise and the other in the opposite direction (counter-clokwise). In this way, they load the musculoskeletal system symmetrically.

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
