# Peer review of "Right Knee—The Weakest Point of the Best Ultramarathon Runners of the World? A Case Study"

_ijerph, 2020, doi:10.3390/ijerph17165955_

Round 1

Reviewer 1 Report

The authors present a case Report on an Ultrmarathoner suffering from knee pain after the run and specific alterations in sofft tissues as desribed by an immediatly performed MRI scan. Three years after the tissue alterations (with adequateley discussed exception of a miniscus tear) were not detectable anymore. Furthermore, they asked another 10 runners for comparable problems.

The mansucript ist well and strictly forward written. I have only minor concerns mainly dealing with the title of the manuscript itself. The titel " Right Knee—The Weakest Point of the Best Ultramarathon Runners of the World?" might lead to the impression of a proper performed study rather than a case report.

Author Response

Reviewer 1

Comments and Suggestions for Authors

The authors present a case Report on an Ultrmarathoner suffering from knee pain after the run and specific alterations in sofft tissues as desribed by an immediatly performed MRI scan. Three years after the tissue alterations (with adequateley discussed exception of a miniscus tear) were not detectable anymore. Furthermore, they asked another 10 runners for comparable problems.

The mansucript ist well and strictly forward written. I have only minor concerns mainly dealing with the title of the manuscript itself. The titel " Right Knee—The Weakest Point of the Best Ultramarathon Runners of the World?" might lead to the impression of a proper performed study rather than a case report. 

Answer to the Reviewer 1:

Dear Reviewer 1

Thank you very much for the time devoted to the analysis of the work and for its high evaluation. The work is written and correctly qualified as a case study. The MRI examination was performed on only one athlete. As you noticed, out of ten other UM runners questioned about, inter alia, pain after the run, 5 also reported pain in the right knee. Among the surveyed competitors it is impossible not to notice one ultra-marathon runner - a woman- who in this run set a phenomenal world record in 24H UM with a score of 256,245 km. (lines 19-21: Different kinds of overloading of the right lower limb, including right knee pain (5 athletes), were indicated by 6 out of 10 competitors from the top 20 in the run, including a woman setting the world record .” The title of the work is intended to highlight this circumstance as well. This one race was attended by two of the world's top 24UM runners who finished the race with the same clinical symptoms (pain in the right knee). I trust that this explanation of the title of the work "provoking" to its in-depth analysis and this commentary will satisfy you. Regardless, I changed to the title ( the title now: Right Knee—The Weakest Point of the Best Ultramarathon Runners of the World? – A Case Study )

Yours sincerely

Robert Gajda

Reviewer 2 Report

Thank you for the opportunity to review the manuscript titled “Right knee - the weakest point of the best Ultramarathon runners of the world?”. This case report is well written and clearly presented. I only offer a few comments for the authors consideration.

  1. I found the abstract to be unnecessarily lengthy and containing far too much information. I strongly recommend reducing the length to approximately half of the current length.
  2. Although I agree with the authors that the right knee pain is likely to be related to overloading caused by excessive clockwise loop running, such causation cannot be inferred from the present research. Case reports are unable to determine causation. The authors should therefore replace all language that infers causation with more cautiously phrased non-causal language.
  3. I suggest that the authors add a short paragraph at the outset of the Results section that briefly describes the athlete’s performance (e.g. distance, number of corners) and onset and post-race presentation of knee pain (i.e. non-MRI findings).

Author Response

Reviewer 2

Thank you for the opportunity to review the manuscript titled “Right knee - the weakest point of the best Ultramarathon runners of the world?”. This case report is well written and clearly presented. I only offer a few comments for the authors consideration.

  • I found the abstract to be unnecessarily lengthy and containing far too much information. I strongly recommend reducing the length to approximately half of the current length.
  • Although I agree with the authors that the right knee pain is likely to be related to overloading caused by excessive clockwise loop running, such causation cannot be inferred from the present research. Case reports are unable to determine causation. The authors should therefore replace all language that infers causation with more cautiously phrased non-causal language.
  • I suggest that the authors add a short paragraph at the outset of the Results section that briefly describes the athlete’s performance (e.g. distance, number of corners) and onset and post-race presentation of knee pain (i.e. non-MRI findings).

Answers for the Reviewer 2:

Dear Reviewer

Thank you very much for the time devoted to the analysis of the work and for its high evaluation.

Ad 1)

 I reduced  the length of the abstract deeply ( lines:12-30)

Answer for the reviewer– continuation

Ad 2)

Although I agree with the authors that the right knee pain is likely to be related to overloading caused by excessive clockwise loop running, such causation cannot be inferred from the present research. Case reports are unable to determine causation. The authors should therefore replace all language that infers causation with more cautiously phrased non-causal language.

Answer: I changed language : ,,The affected competitors indicated suggested as cause for discomfort the shape of the loop and running direction” ( line 21).

Results indicated can suggest, that a one-way, clockwise, 24-h UM run had an adverse effect on the athlete's right knee as a result of unsymmetrical load and distorted his performance (line 25).

It is highly likely that the lateral knee compartment overload caused by completion of over 5 hundred rectangular corners was could be the reason for the described MRI observations. Most changes (except for the lateral meniscus tear, which cannot heal) were not observed after 3 years.(lines: 187-189)

Our results indicate suggest that a one-way, clockwise, 24-h UM run had an adverse effect on the athlete's right knee as a result of unsymmetrical load which distorted the final result ( lines: 271-272)

Ad 3)

I suggest that the authors add a short paragraph at the outset of the Results section that briefly describes the athlete’s performance (e.g. distance, number of corners) and onset and post-race presentation of knee pain (i.e. non-MRI findings).

Answer: done : lines 133-137 ( below)

3.1 Data related to 24-h UM run

,,The runner during 24-h UM ran 258.228 km. He started UM without any symptoms ( no pain).  The athlete started to experienced right knee pain, which appeared after about 12 hours of running (approximately 130 km,) and intensified up to the end. He ran on a 1984-m rectangle-loop (950x42m). The winner completed together 516 rectangular corners running in an atypical clockwise fashion Women's world record in the 24H UM was set in the same race with a result of 256.245 km. ”

Yours sincerely

Robert Gajda

Reviewer 3 Report

The authors state that there are not many reports on the impact of UM on athletes' knees, and none describing the impact of running direction and loop shape.However, it is not clear the necessity and the impact of this study. Accordingly, I suggest to develop their Introduction again. Also, it would be useful to state at the title that this is a Case report.

At the methods section the authors should mention the environmental conditions when the athlete run the Marathon.

Also the discussion and the perspectives section should be rewritten. The authors should discuss their findings and compare them with those of other similar studies. 

Author Response

Reviewer 3

Comments and Suggestions for Authors

  • The authors state that there are not many reports on the impact of UM on athletes' knees, and none describing the impact of running direction and loop shape. However, it is not clear the necessity and the impact of this study. Accordingly, I suggest to develop their Introduction again.
  • Also, it would be useful to state at the title that this is a Case report.

  • At the methods section the authors should mention the environmental conditions when the athlete run the Marathon.

  • Also the discussion and the perspectives section should be rewritten. The authors should discuss their findings and compare them with those of other similar studies.

Answer for the Reviewer 3

Dear Reviewer 3

Thank you very much for the time devoted to the analysis of the work for its evaluation. Below all explanations and corrections. I did what I could to take into account your suggestions regarding the article, while not significantly changing the article in the elements positively assessed by the other two reviewers. I hope that the introduced changes will satisfy your expectations.

Regards

Robert Gajda

Ad 1) The authors state that there are not many reports on the impact of UM on athletes' knees, and none describing the impact of running direction and loop shape. However, it is not clear the necessity and the impact of this study. Accordingly, I suggest to develop their Introduction again.

 Answer: I developed the Introduction again trying to clarify the necessity and the impact of the study ( lines: 36-49).

Ultramarathon (UM) races (distance longer than 41,195 meters) are a significant challenge for the competitors' organism. Preparations for these races take years, and often crown a runner's career. Overload injuries of the musculoskeletal system due to gruelling training is highly probable, and often results in the inability to start the race for which the athlete has prepared for years. It is not uncommon for an overload injury to end a runner’s career. Competitors can train to minimize the probability of injury using various training stimuli, rehabilitation, or avoiding repetitive movements. This special attention to training may be insufficient for the safe completion of a 24-hour (24H) race if it takes place, for example, on a rectangular loop and in one direction. This was the case during the 24H race in the Polish Championships. Six of the top 10 competitors reported right knee pain, probably due to taking several hundred rectangular turns to the right. Two of the world's leading UM runners participated in the run, and both experienced similar pain. Fortunately, their overload injuries did not end their careers. However, such a situation requires an assessment of the impact of such organized races on the competitors’ health, and indicates the necessity and impact of this study.

Ad 2)

Also, it would be useful to state at the title that this is a Case report.

Answer:

.The work is written and correctly qualified as a case study. The MRI examination was performed on only one athlete. As you noticed, out of ten other UM runners questioned about, inter alia, pain after the run, 5 also reported pain in the right knee. Among the surveyed competitors it is impossible not to notice one ultra-marathon runner - a woman- who in this run set a phenomenal world record in 24H UM with a score of 256,245 km. (,, ... 26-28 lines: Different kinds of overloading of the right lower limb, including right knee pain (5 athletes), were indicated by 6 out of 10 competitors from the top 20 in the run, including a woman setting the world record in her category (256.245 km). The title of the work is intended to highlight this circumstance as well. This one race was attended by two of the world's top 24UM runners who finished the race with the same clinical symptoms (pain in the right knee). I trust that this explanation of the title of the work "provoking" to its in-depth analysis and this commentary will satisfy you. Regardless, I changed the title giving information about type of article: Right Knee—The Weakest Point of the Best Ultramarathon Runners of the World? –A Case Study.

Ad 3)

At the methods section the authors should mention the environmental conditions when the athlete run the Marathon.

Answer:

I added section about weather condition in form of figure 2B ( lines:109-115)

Ad 4)

Also the discussion and the perspectives section should be rewritten. The authors should discuss their findings and compare them with those of other similar studies.

Answer:

In the section 4.1 Discussion of the Results – we discussed the result ( lines: 173-207). Then , in the section : 4.2. Knee overload injuries in UM- runners, ( lines: 208-225) we discussed injuries of the knee in different ultramarathons. Unfortunately, we cannot relate to similar studies because our research was innovative in this area and the effects of ultramarathon running on a single-line loop on the knee joints have never been described before. On the other hand, we made extensive reference to the available literature describing the effects of UM on the knee joints of runners.

I think, perspectives (4.5. Perspectives

To properly document the harmful effects of running in one direction in a 24-hour UM run, MRI tests of both knee joints on many competitors should be performed both before and after the run. Ideally, the study should be conducted in three different 24-hour UMs conducted in clockwise, counter-clockwise and alternating directions. Such results could then provide a strong argument for the competition organizers of 24-hour UM to consider loop characteristics and competition methodology.) clearly indicate what kind of research should be performed in the future and what should be done with the results ( lines: 264-269)

Round 2

Reviewer 3 Report

I was happy to see that the author made all the suggested corrections. I suggest publication.